# Validity and Reliability of a Tool for Accelerometric Assessment of Balance in Scholar Children

**DOI:** 10.3390/jcm10010137

**Published:** 2021-01-03

**Authors:** Jesús García-Liñeira, Raquel Leirós-Rodríguez, Vicente Romo-Pérez, Jose L. García-Soidán

**Affiliations:** 1Faculty of Education and Sport Sciences, Universidade de Vigo, Campus a Xunqueira, s/n, 36005 Pontevedra, Spain; jesgarcia@alumnos.uvigo.es (J.G.-L.); vicente@uvigo.es (V.R.-P.); jlsoidan@uvigo.es (J.L.G.-S.); 2Nursing and Physical Therapy Department, Faculty of Health Sciences, Universidad de León, Ave. Astorga, 15, 24401 Ponferrada, Spain

**Keywords:** accelerometer, biomechanical phenomena, gait analysis, kinetics, postural balance

## Abstract

In Pediatrics, balance is assessed through low-sensitivity clinical tests which identify developmental alterations at already advanced stages that cannot be detected at earlier stages. Therefore, the aim of this study was to develop an easily applicable quantitative tool that can be used to evaluate postural control. Consequently, a cross-sectional study was carried out with 91 healthy children. All of them performed a series of six accelerometric functional tests and four clinical tests of balance (Modified Flamingo Test, Bar Test, Babinski–Weil Test, and Fukuda Stepping Test). The Bar Test obtained mild inverse correlations with accelerations produced in the mediolateral axis and the root mean square of all the tests in monopodal support. The Flamingo Test obtained direct correlations with the root mean square of the tests in monopodal support and with the mediolateral axis of the monopodal tests and gait. The pediatric balance assessment scale consists of three factors and eleven items extracted from five accelerometric functional tests: the monopodal balance test with six items, normal gait test with three items, and bipodal balance test with two items. This tool is easy to apply and allows analysis in the evaluation of the balance state based on the accelerations of the center of mass.

## 1. Introduction

Postural control is the capacity of an individual to keep his/her center of gravity (or center of mass, CM), whose location varies depending on the posture, over his/her base of support; either remaining static or during the realization of other simultaneous movements. This allows the person to maintain the desired position against gravity and interact with the environment [1]. Postural control is a system that employs and coordinates different strategies: biomechanical strategies (especially in the lower limbs), movement strategies (through unconscious postural adjustments), sensory strategies (using somatosensory, visual and vestibular information) and cognitive strategies (through attention and learning from experiences) [2,3]. In turn, body stability and balance are maintained when the imaginary vertical line that connects the CM to the floor is within the limits of the support area; thus, the larger the support area, the easier it is to remain stable in balance [4]. Consequently, this should be the study object to evaluate balance, in contrast to the traditional way, which is focused on the center of pressures. The latter is strongly related to postural control, although it is only an indirect measurement of it; moreover, it is only valid if the body is considered as an inverted pendulum and not as a multisegment system, which is closer to reality [5,6].

In pediatrics, balance is assessed through low-sensitivity clinical tests, such as the Babinski–Weil Test and the Fukuda Stepping Test, which identify developmental alterations at already advanced stages of deterioration that cannot be detected in earlier stages [7,8,9]. Following the implementation of previous studies in adult and older populations [10,11], the use of three-dimensional accelerometers should be generalized to assess postural control in children, providing healthcare professionals and experts in movement and psychomotricity with a quantitative, sensitive and reliable tool that is easy to use and interpret. In fact, a recent systematic review identified accelerometers as instruments that provide reliable information about postural control in a more sensitive manner than clinical test sets traditionally used in the clinical scope. However, accelerometry has a discrete degree of implementation as an evaluation tool to assess balance and its method has not yet been protocolized and standardized [12].

Therefore, the aim of this study was to determine the relationship between tests for the clinical assessment of balance in children and the items that explain each of them, using accelerometric variables, to develop an easily applicable quantitative tool that can be used to evaluate postural control.

## 2. Materials and Methods

### 2.1. Study Design and Sample

This descriptive, cross-sectional study was performed using a convenience sample of children from Pontevedra (Spain). All participants were recruited from their school. Participants who met any of the following exclusion criteria were unable to participate: children who were unable to walk independently or without external orthotics, those who could not stand for 60 s or more, children with any specific contraindications to the evaluation tests; or children who did not provide parental informed consent to participate in the study.

This study involved 91 healthy children (50.5% boys) with a mean age of 9 ± 1.8 years. The selection process had the objective of including five girls and five boys of each age between 6 and 12 years (finally, there were 13 participants of each age). For each participant, measures of weight and height were taken, from which their body mass index (BMI) was calculated. 

### 2.2. Instrument and Processing of Data

The accelerometer GT3X+ (Actigraph^®^, Pensacola, FL, USA) provides data on body movements in three axes: the Pensacola measurements were configured for a time frame of 1 s. This protocol has previously been applied and validated in both adults and children [13,14].

Using a low frequency helps to eliminate any acceleration noise [6]. Thus, a frequency of 50 Hz was selected to achieve greater accuracy in the analysis of postural balance [13]. Raw data were collected at the selected sample rate and processed using ActiLife software (Actigraph^®^, Pensacola, FL, USA). The accelerometers used in the tests were synchronized using a PC and ActiLife software.

### 2.3. Procedure

The first step of the procedure was to explain the purpose of the study to the participants and their parents, and give them a brief description of what they were supposed to do. The parents of all participants signed an informed consent form in accordance with the Declaration of Helsinki (revised 2013). All the procedures performed in the studies involving human participants were in accordance with the ethical standards of the Commission of Ethics of the Faculty of Sciences of Education and Sport of the University of Vigo (Spain; number 3-0406-14).

Once the informed consent form was signed, the participant’s data (full name and age) were collected. Then, the anthropometric measurements (weight and height) were obtained using a scale (SECA^®^, Berlin, Germany) and a stadiometer (SECA^®^, Berlin, Germany). For both anthropometric measurements, the students were asked to remove footwear and any unnecessary clothing.

After these measurements, the accelerometer was placed in the lumbar medial zone (coincident with the fourth lumbar vertebra). The specific location was the 4th lumbar vertebra that has been demonstrated to reflect the behavior of the CM [15]. The devices were attached with adhesive tape to avoid displacement. The trial was explained to the participants, and they were accompanied to a measurement room (an empty classroom) for testing.

In order to carry out measurements of the accelerations of the CM, each participant completed six trials, each of which were repeated three times. The tests were separated by intervals of 30 s to prevent the effects of lower limb muscle fatigue [16]. The following trials were performed: (a) bipodal standing balance with eyes open (BOE); (b) bipodal standing balance with eyes closed (BCE); (c) balancing on one leg with eyes closed (OLCE); (d) balancing on one leg with eyes open (OLOE); (e) dynamic balancing on one leg on a foam mat with eyes open to induce the onset of dynamic equilibrium reactions (DOL); and (f) normal gait to a cone located 10 m away, with each participant walking around the cone and returning to the starting point (NG). Unstable surface tests were performed on a mat with a density of 30 kg/m^3^. Participants were told that if they suffered an imbalance while in a monopodal stance that required them to use their other leg for support, they should attempt to recover the requested position in the shortest time possible. During testing the subjects wore socks and easy clothing so that they could perform the tests comfortably. All participants were instructed to separate the feet at hip height in bipodal balance tests. In the monopodal balance tests, the participants chose the leg on which to make the support. To do this, they were allowed to make initial attempts to make the selection (which they had to respect for all the tests).

The sequence of the trials was determined taking into account possible fatigue of the lower limbs. The trial order was OLCE–NG–BOC–OLOE–BOE–DOL–BOC–OLCE–BOE–DOL–NG–OLOE–BOC–DOL–BOE–OLOE–OLCE–NG, with each trial (BOC, BOE, OLCE, OLOE, DOL and NG) performed three times.

### 2.4. Clinical Indicators 

The clinical balance tests were performed on the same day and in a completely silent environment to avoid distractions or interference [17]. All tests were performed with participants wearing socks, without footwear and easy clothing, allowing them to perform the tests comfortably. These tests were: (a)Modified Flamingo Test (FT): Static balance was assessed by this test adapted from the Flamingo Test for use in children [18]. The children were asked to stand barefoot on one leg with eyes closed for 30 s. It was carried out three times to subsequently average the three scores obtained with both hands at the waist at the level of the iliac crest. The test score was the number of floor touches with a free foot or eye openings during 30 s, higher number of floor touches and eye openings indicating poorer static balance [19,20].(b)Bar Test (BT): Dynamic balance during tandem gait was assessed by this test. For this, the subject had to be barefoot and standing on a bar 2 m long and 5 cm wide and between 30 and 40 cm high (an inverted Swedish bench can be used for this purpose). The boy or girl had to walk on the bar to the mark located at 2 m, to be able to turn and continue walking again. They had to do this round trip as many times as possible, for a period of 45 s, until they lost their balance. The measurement was established based on the number of meters that the child was able to travel, over 45 s. This test was explained to the participants and they were allowed to test it before making the three measurements after which the average of the results obtained was found. This test presents reliability results of 0.98 for 12 year old subjects [21,22].(c)Babinski–Weil Test (BWT): This test evaluates the integrity of the vestibular system. It consists of gait forwards and backwards with eyes closed. Subjects stand upright and are asked to walk forward six steps in a straight line and backward the same six steps, retracing the above steps. This procedure was repeated three times (36 steps total) with their arms resting comfortably at their sides. If the subject performs the walk in a straight line, the test is suitable; if there are lateral deviations of 45º or more it is not suitable [23,24].(d)Fukuda Stepping Test or Unterberger–Fukuda Test (FST): also called the pretended gait test is used to assess dynamic balance. It consists of carrying out gait on the site with eyes closed, raising the knees up to 90°, and simulating a military gait, with a frequency of 50 steps per minute with arms resting comfortably at their sides. It is a clinical test evaluated as suitable or not, by checking if there is a rotation to the right or to the left of more than 45° [23].

### 2.5. Instrument

To develop the accelerometric assessment tool, it was first decided which variables were to be included. The procedure was performed as follows: (1) a review of the literature on pediatric balance assessment [12] to show us which accelerometric tests were relevant (bipodal and single-leg balance with eyes open, closed, monopodal balance on a mat with eyes open and normal gait [13,14]) as well as the most important clinical tests for the purpose (FT, BT, BWT, and FST); (2) a clinical and accelerometric assessment of the sample; (3) an exploratory factor analysis (EFA).

### 2.6. Statistical Analysis

A descriptive analysis of all the study variables was performed through calculation of the average values (to determine the central tendency) and the standard deviation (as a measure of dispersion).

The variables showed a normal distribution according to the Kolgomorov–Smirnov test (*p* > 0.05), and there was homogeneity of variances, applying the Levene test. The *t*-test was used to verify the existence of significant differences between the sexes, for each of the continuous variables analyzed.

The Pearson correlation coefficient was performed between the clinical indicators (FT, BT, BWT and FST) and the accelerometric variables (the mean values of the three axes and the RMS for each accelerometric functional test) to find the relationship between them. The correlation between the different accelerometric variables was also found, to determine the relationship between the accelerations produced and the different balance tests performed.

We created linear regression models using the accelerometric mean outcomes (independent variables: mean values of the three axes and the RMS for each accelerometric functional test) as well as clinical indicators (dependent variable: FT, BT, BWT, and FST), along with adjustments for age for the data of all participants. 

For statistical analysis, an EFA of the principal components with varimax rotation was performed to determine the factorial structure of the tool and which items were relevant (for its load factor), eliminating those which had lower loads than 0.85 (72.25% of the variance), thereby sanctioning the reliability of the scale. To test internal consistency a reliability analysis was also performed by calculating Cronbach’s alpha of each factor and of the total scale. 

The program STATA V.13 (State Corp., College Station, TX, USA) was used for statistical analysis.

## 3. Results

### 3.1. Descriptive Analysis

Regarding the clinical indicators, we observed differences between males and females in the FT and FST results. The FT results were significantly worse in boys, with an average of 4.5 floor taps per attempt compared to the 2.7 floor taps obtained by the girls. In contrast, the FST showed 89.4% of valid results in the group of boys compared to 4.7% obtained for the group of girls. The results of the BT and BWT were similar for both boys and girls. For the entire sample, the BT results showed an average of 5.5 meters, and 44.9% of the participants conducted the BWT successfully.

With regard to the accelerometric results, there were no significant differences between sexes in any of the analyzed axes in the BOC test. However, the BOE results showed differences in the results obtained for the anteroposterior axis and the RMS; the OLOE results were different in these two variables and in the mediolateral axis. In the OLCE and DOL, the two sexes differed for the accelerometric results of the mediolateral axis and RMS. In all the described cases, the differences were obtained through accelerometric results, which were significantly higher (over 50%) in the group of boys in several variables (Table 1). Lastly, in the NG test, both sexes obtained significantly different results in the vertical axis, anteroposterior axis, and RMS. In the NG, it was the girls who obtained significantly higher results, except for the mediolateral axis, in which both groups obtained the same average accelerometric value.

### 3.2. Correlation Analysis

The 24 variables extracted from the accelerometric analysis and the clinical indicators were analyzed to determine the possible correlations between them. The BT obtained mild inverse correlations (r =−0.4; *p* < 0.001) with the accelerations produced in the mediolateral axis and the RMS of all the tests conducted in monopodal support (OLCE, OLOE, and DOL). The FT obtained direct mild correlations with the RMS of the three tests conducted in monopodal support (r = 0.6; *p* < 0.001) and with the accelerations in the mediolateral axis of the OLCE, OLOE and NG tests (r = 0.7; *p* < 0.001). The accelerometric variables and clinical indicators were not correlated with BMI, body weight or height.

The high correlations found between the different accelerometric tests were between mediolateral axis and the RMS for the DOL test and the accelerations in the mediolateral axis and the RMS for the OLCE and OLOE tests (r > 0.9; *p* < 0.001; for the eight correlations). The results obtained in the mediolateral axis and the RMS in the OLOE and OLCE tests were also highly correlated (r > 0.9; *p* < 0.001; for the four correlations).

### 3.3. Linear Regression Analysis

The linear regression analysis demonstrated the influence of the clinical indicators on the accelerations of the body (Table 2). The RMS during the monopodal tests (OLOE, OLCE, and DOL) were related with the results of BT. The FT were strongly related to the RMS of the static balance tests with open eyes (BOE, OLOE, and DOL). Of them all, the RMS of the BOE test was the accelerometric variable that showed stronger relationships with the FT (β = 1.49; *p* < 0.01).

### 3.4. Factor Analysis

Initially the scale has six factors (one for each accelerometric test performed) and a total of 24 items for the scale (the mean values of the three axes and the RMS for each test). Then, three factors with a combined total of 11 items appeared (Table 3). This analysis discarded the initial 13 items that were part of the assessment. These items were eliminated as they had low levels of reliability, with less than 0.85 factor loadings. The resulting items were grouped into three factors: (1) monopodal balance (MB), with six items; (2) normal gait (NG), with three items; and (3) bipodal balance (BB), with two items.

The internal consistency of the scale was estimated with Cronbach’s alpha coefficient: (MB: 0.796; NG: 0.803; BB: 0.852; full scale: 0.808).

## 4. Discussion

The aim of this study was to determine the relationship between the tests for the clinical assessment of balance in children and the accelerometric evaluation of balance and gait. Considering the obtained results, the clinical tests for balance evaluation used in pediatrics are related to accelerometric assessment, although they do not provide comparable results, since they do not produce exactly the same information, while the instruments they employ have different sensitivity.

The results obtained in this study regarding the state of postural control are directly influenced by the characteristics of the sample used: healthy boys and girls with normal body weight and without neurological pathologies or developmental alterations. In the children included in this study, no relationship was observed between body weight or BMI and the result of any of the clinical indicators or accelerometric variables, which is in contrast to the results obtained in previous studies conducted with adolescents and pre-adolescents [25,26]. It has been demonstrated that obesity modifies the geometry of the body, increasing the weight and volume of the limbs and, consequently, the predisposition to falling, due to biomechanical limitations [26,27]. Therefore, this controversial finding may be due to the young age of the study sample, or the lack of overweight and obese participants who could confirm the existence of such a relationship between BMI and postural control during childhood.

In the present study, we identified a better result in the balance tests (both clinical and accelerometric) for the girls with respect to the boys. This finding is in line with the results of previous studies that reported better postural stability in girls compared to boys [25,28,29]. In girls, the maturation of the vestibular system and of the areas of the brain responsible for postural control takes place at an earlier age, whereas boys tackle each sensory input related to postural control separately and rely to a greater extent on somatosensory feedback [28,29,30,31]. This explains why, in this study, there were more differences between sexes in the accelerometric evaluation tests on pads than in the tests with eyes closed.

Regarding the relationship between the clinical indicators and the accelerometric variables, the number of meters covered in the BT was inversely correlated with the accelerations in the mediolateral axis and the RMS for the tests conducted in monopodal support. However, the regression analysis confirmed that the variables that were most strongly correlated with such clinical tests were the RMS of the tests in monopodal support. This finding is justifiable, since BT consists of tandem gait, which challenges the participant to walk in more difficult support conditions and prolongs the duration of the gait phases, involving monopodal support [32]. Likewise, FT was related to the same accelerometric variables as BT and to the accelerations in the mediolateral axis during the gait. Of these relationships, it is worth highlighting the predominant role of the accelerations in the mediolateral axis during the postural control system challenging tests. This is in contrast to the findings reported in adult and older populations, in which the straightening and balancing reactions, although also observed to a great extent in the mediolateral axis, occurred predominantly in the anteroposterior axis [6].

With respect to the correlation between the different accelerometric variables, there was a strong relationship between the results obtained in the mediolateral axis and the RMS of the OLOE and OLCE tests, and between the results of the latter two with those of the DOL test. This is justified by the fact that the three tests are conducted in monopodal support. However, considering the strong correlation between them and taking into account that they could be providing repetitive information, this must be addressed in future studies in order to determine the most appropriate test for the evaluation of balance in children.

Furthermore, this study was also aimed at identifying the representative factors of the balance construct and the items that explain each of them, using a model based on accelerometric variables, with the aim of developing a simple, objective and quantitative tool to evaluate balance, and which can be easily used in the clinical context. The highest score was obtained in the elements of the representative factor of monopodal balance, where the participant is challenged to keep postural control while standing on one leg, both on the floor and on a pad, with eyes opened and closed. Then, the representative factor of gait was defined and, finally, the lowest score corresponded to the items of bipodal balance, which included two accelerometric variables extracted from the test in bipodal support with eyes closed. The three factors included elements related to the RMS and complemented them with items related to either the mediolateral axis (in the factors of bipodal and monopodal balance) or the vertical and anteroposterior axes (in the gait factor). The three factors show a good internal consistency, with values above 0.88, which explains over 74.8% of the variance, in the case of the two items with lower factor loading (accelerations in the vertical and anteroposterior axes during gait). Therefore, all the variables included in this accelerometric assessment instrument are considered to be transcendental, since each of them explains differentiated and complementary aspects of the subsystems, which are involved in postural control and in its complex and multidimensional nature. The final combination of the five tests (all the tests used initially, except BOE) can be used to quantify the capacity of a child to keep an upright position against gravity, in different support, surface, and vision conditions. Consequently, it allows identifying alterations of structures of the nervous system in psychomotor development and maturation, such as the peripheral nervous system, the somatosensory system, and the vestibular system. The system to validate the pediatric assessment of balance, using the accelerometry methods presented in this study, can represent quantitatively the state of postural control of healthy boys and girls through the selection of its items based on psychometry: a short scale of five tests and 11 items, extracting only two variables out of four of the tests and three variables of the remaining test (NG). Lastly, it is important to highlight the congruence between the initial correlation and linear regression analyses and the final design of the assessment instrument. In all cases, the accelerations in the mediolateral axis and the RMS during the balance tests were the variables that provided the most information about postural control, which was confirmed through the strong relationship between their magnitude and the results of the clinical indicators. However, the tool includes the accelerations obtained in the anteroposterior and vertical axes and the RMS of the NG, which were not related in any of the statistical analyses conducted with the clinical indicators. This could be due to the fact that the clinical tests performed in this study are all representative of the static balance state, whereas gait is a much more complex task that involves aptitudes that are not challenged by such clinical tests.

This study has several limitations. First, due to the small sample size, the results have poor generalizability. The second limitation is the lack of longitudinal data to evaluate the development and maturation of the postural control system. Lastly, this tool does not allow calculation of the risk of falling nor diagnosing pathologies in a direct manner, alth ough it does allow identification of the accelerometric variables that quantify the balance state throughout childhood.

Future studies must expand these findings to larger populations with the aim of establishing reference accelerometric values and their percentiles, in order to standardize and generalize the use of this assessment method and, at the same time, apply this method in populations of children with vestibular, visual, neurological, and/or developmental pathologies. This would help in the early identification of such diseases and increase the sensitivity of the evaluation and follow-up processes, consequently improving the pharmacological and physiotherapeutic treatments they receive.

## 5. Conclusions

The pediatric balance assessment scale presented in this study consists of three factors and eleven items that were extracted after conducting five accelerometric functional tests. This tool obtained good validity, is easy to apply and allows the analysis of the balance state based on the accelerations of the CM. This will enable the objective and precise identification of inadequate states of postural control functionality, along with the introduction of this tool in rehabilitation and physical activity programs specifically aimed at stimulating and developing physical aptitudes related to balance and postural control.

## Figures and Tables

**Table 1 jcm-10-00137-t001:** Descriptive statistics of accelerometric and clinical variables (mean ± standard deviation).

Variable	All (*n* = 91)	Male (*n* = 46)	Female (*n* = 45)
Age (years)	9.1 ± 1.9	9.1 ± 2	9 ± 1.8
Height (cm)	138.2 ± 12.2	137.7 ± 12	138.8 ± 12.6
Weight (kg)	37.3 ± 11.3	37.1 ± 10.8	37.5 ± 12
Body Mass Index (kg/m^2^)	19.1 ± 3.6	19.2 ± 3.6	19.1 ± 3.5
Clinical Indicators			
Flamenco test (*n*)	3.6 ± 0.5	4.5 ± 0.7 *	2.7 ± 0.7 *
Bar test (m)	5.5 ± 0.4	5.4 ± 0.6	5.7 ± 0.6
Babinsky–Weil Test (dictomic)			
Valid (*n*)	40 (44.9%)	21 (45.7%)	19 (44.2%)
Fail (*n*)	49 (55.1%)	25 (54.4%)	24 (55.8%)
Fukuda Stepping Test (dictomic)			
Valid (*n*)	7 (7.8%)	42 (89.4%) **	2 (4.7%) **
Fail (*n*)	83 (92.2%)	5 (10.6%) **	41 (95.4%) **
Accelerometric variables			
Bipodal balance with eyes open (g)			
Vertical axis	0.0 ± 0.1	0.0 ± 0.1	0.0 ± 0.0
Medio-lateral axis	0.2 ± 0.6	0.3 ± 0.8	0.1 ± 0.2
Antero-posterior axis	0.2 ± 0.4	0.2 ± 0.5 *	0.1 ± 0.2 *
Root Mean Square	0.3 ± 0.8	0.5 ± 1 **	0.1 ± 0.3 **
Bipodal balance with eyes closed (g)			
Vertical axis	0 ± 0.1	0 ± 0	0 ± 0.1
Medio-lateral axis	0.1 ± 0.3	0.1 ± 0.4	0 ± 0.2
Antero-posterior axis	0.1 ± 0.2	0.1 ± 0.2	0 ± 0.1
Root Mean Square	0.1 ± 0.4	0.2 ± 0.5	0.1 ± 0.3
Monopodal balance with eyes open (g)			
Vertical axis	3.8 ± 6.9	4.9 ± 7.8	2.6 ± 5.7
Medio-lateral axis	6.7 ± 8.7	8.5 ± 9.8 *	4.7 ± 6.9 *
Antero-posterior axis	5 ± 6.8	6.4 ± 7.3 *	3.5 ± 6 *
Root Mean Square	10.9 ± 14.4	13.9 ± 16 *	7.7 ± 11.8 *
Monopodal balance with eyes closed (g)			
Vertical axis	7.5 ± 11	9.4 ± 11.2	5.4 ± 10.5
Medio-lateral axis	12.5 ± 11.4	15 ± 11.1 *	9.6 ± 11.1 *
Antero-posterior axis	8 ± 9	9.4 ± 9	6.5 ± 9
Root Mean Square	19.6 ± 19.5	23.5 ± 19.3 *	15.3 ± 19 *
Monopodal balance on mat with eyes open (g)			
Vertical axis	8.6 ± 14.3	11 ± 15.6	6.1 ± 12.3
Medio-lateral axis	11.7 ± 13.5	14.5 ± 14.4 *	8.7 ± 11.8 *
Antero-posterior axis	8.2 ± 11	9.9 ± 10.6	6.4 ± 11.3
Root Mean Square	19.7 ± 23.9	24.3 ± 25.4 *	14.7 ± 21.3 *
Normal gait (g)			
Vertical axis	37.8 ± 15.1	34.4 ± 15.5 *	41.5 ± 13.9 *
Medio-lateral axis	20.3 ± 8.4	20.3 ± 8.4	20.3 ± 8.5
Antero-posterior axis	29.6 ± 8.5	27.1 ± 8.5 **	32.5 ± 7.7 **
Root Mean Square	56.5 ± 16.6	52.6 ± 17 *	60.7 ± 15.3 *

*t*-test between sex groups: * *p* < 0.05; ** *p* < 0.01.

**Table 2 jcm-10-00137-t002:** Linear regression models for the Clinical Indicators (continuous variables).

Variable	Bar Test			Flamingo Test		
	β	SE	95% CI	β	SE	95% CI
BOE_RMS_	−0.67	0.52	−1.69–0.36	1.49 **	0.52	0.46–2.52
BCE_RMS_	0.26	0.96	−1.65–2.17	−1.26	0.99	−3.23–0.71
OLOE_RMS_	−0.07 *	0.03	−0.13–−0.01	0.15 ***	0.03	0.09–0.2
OLCE_RMS_	−0.05 *	0.02	−0.1–−0.00	0.11	0.02	0.06–0.15
DOL_RMS_	−0.04 *	0.02	−0.08–−0.00	0.06 **	0.02	0.02–0.11
NG_RMS_	0.03	0.02	−0.02–0.08	−0.01	0.03	−0.06–0.04

β: regression coefficient; SE: standard error; 95% CI: 95% confidence interval; BOE_RMS_: root mean square during standing balance with eyes open; BCE_RMS_: root mean square during standing balance with eyes closed; OLOE_RMS_: root mean square during monopodal balance with eyes open; OLCE_RMS_: root mean square during monopodal balance with eyes closed; DOL_RMS_: root mean square during monopodal balance on mat with eyes open; NG_RMS_: root mean square during normal gait. * *p* < 0,05; ** *p* < 0.01; *** *p* < 0.001.

**Table 3 jcm-10-00137-t003:** Exploratory factor analysis.

Accelerometric Variable	Factor 1 (MB)	Factor 2 (NG)	Factor 3 (BB)
Medio-lateral axis during monopodal balance on mat	0.968		
Root mean square during monopodal balance on mat	0.964		
Medio-lateral axis during monopodal balance with eyes open	0.956		
Root mean square during monopodal balance with eyes open	0.952		
Root mean square during monopodal balance with eyes closed	0.94		
Medio-lateral axis during monopodal balance with eyes closed	0.938		
Root mean square during normal gait		0.975	
Vertical axis during normal gait		0.887	
Antero-posterior axis during normal gait		0.882	
Medio-lateral axis during standing balance with eyes closed			0.928
Root mean square during standing balance with eyes closed			0.92

MB: monopodal balance; NG: normal gait; BB: bipodal balance.

## Data Availability

The data presented in this study are available on request from the corresponding author.

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
