# Peer review of "Validity and Reliability of a Tool for Accelerometric Assessment of Balance in Scholar Children"

_jcm, 2021, doi:10.3390/jcm10010137_

Round 1

Reviewer 1 Report

Title: “Validity and Reliability of a Tool for Accelerometric Assessment of Balance in Scholar Children

General comments:

The aim of the present study was to develop an easily applicable quantitative tool that can be used to evaluate postural control. 91 healthy children performed a series of accelerometric tests and clinical tests of balance. Authors found that the paediatric balance assessment scale consisted of three factors and eleven items extracted after conducting five functional tests. This tool is easy to apply and allows to analyse in the evaluation of the balance state based on the accelerations of the centre of mass.

Your paper could be of interest for the community but, for my opinion, it needs a major revision before to be suitable for publication. It is not clear what is the purpose of the study, which way you follow to record and analyze data, and finally, which is the tool you suggest to use to assess balance in children.

Abstract:

  • Line 16: the verb is missing: “….. a cross-sectional study with 91 healthy children…”
  • Lines 22-24: conclusions are unclear, you talk about five tests but before mentioned 4 tests. I don’t understand which is the tool, and which are the factors and items….you should rewrite this section to be more precise, the abstract is the first part of your paper that readers will read!!

Introduction:

  • Line 46-48: please discuss the point that bring you to your study hypothesis. Precisely, where you talk about the “implementation of previous studies in adult and older populations the use of three-dimensional accelerometers should be generalized to assess postural control”….what you mean with implementation and generalized. You know that adults are very different from children, so generalization is not appropriate. I suggest to you to discuss why you did your research and which is the connection between Moe-Nilssen 1998 and Leiros-Rodriguez 2020 studies with yours.

Methods:

  • Lines 82-141: I have not understood the connection between the six measurements of the accelerations of the CM (OLCE – NG – BOC – OLOE – BOE – DOL) and the four (not five) tests: Modified Flamingo Test; Bar Test; Babinski-Weil Test; Fukuda Stepping Test. Why you did both?  
  • Paragraph 2.5 does not make sense. What does “the review showed us what accelerometric tests were relevant…” mean? You defined before a lot of measurement and now you don’t know which use?
    • Line 147: “y” is not English
    • Line 148: statistical analysis must be discussed on appropriate paragraph
    • Root means square must be introduced in the appropriate paragraph (2.3) not here.
    • I suggest to you to revise appropriately or delete this paragraph
  • Line 159: please identify between brackets the “clinical indicators” and the “accelerometric variables” for clarity.
  • Lines 163-165: please also here identify variables, insert them between brackets.

Results:

  • Line 183: “anteroposterior axis and the RMS”, what you mean? You did not analyze the RMS of the AP and ML direction? If not, what kind of measure you used for AP and ML?
  • Table 1: what kind of measure is g (accelerometric variables). Moreover, the difference between ALL and ???? for AP what does it means? Did you analyze male vs. female? If yes, why ALL vs. ??? means?

Discussion:

  • Lines 247-254: I suggest to you, to discuss results between male and female, to read and cite the paper of Persiani et al., (2015). Laterality of stance during optic flow stimulation in male and female young adults. BioMed Research International, 2015.
  • Lines 255-267: I suggest to you, to discuss results about gait and posture, to read and cite the paper of Raffi et al., (2017). Angle of gaze and optic flow direction modulate body sway. Journal of Electromyography and Kinesiology, 35, 61-68.

Author Response

Dear Editor and Reviewer of Journal of Clinical Medicine:

Thank you very much for your suggestions and contributions to improve the quality of the manuscript. Following your indications, we respond, point by point, to the reviewers' comments.

In the text, all the modified or added sentences have been written in red to facilitate the correction by the reviewers.

  1. Abstract. Line 16: the verb is missing: “….. a cross-sectional study with 91 healthy children…”

The authors have corrected that sentence.

  1. Abstract. Abstract: Lines 22-24: conclusions are unclear, you talk about five tests but before mentioned 4 tests. I don’t understand which is the tool, and which are the factors and items….you should rewrite this section to be more precise, the abstract is the first part of your paper that readers will read!!

The authors have rewritten the Abstract to improve it and provide all the necessary information for readers.

  1. Introduction. Line 46-48: please discuss the point that bring you to your study hypothesis. Precisely, where you talk about the “implementation of previous studies in adult and older populations the use of three-dimensional accelerometers should be generalized to assess postural control”….what you mean with implementation and generalized. You know that adults are very different from children, so generalization is not appropriate. I suggest to you to discuss why you did your research and which is the connection between Moe-Nilssen 1998 and Leirós-Rodríguez 2020 studies with yours.

The authors have inserted additional information between the sentence to which you refer and the objective of the study.

  1. Methods. Lines 82-141: I have not understood the connection between the six measurements of the accelerations of the CM (OLCE – NG – BOC – OLOE – BOE – DOL) and the four (not five) tests: Modified Flamingo Test; Bar Test; Babinski-Weil Test; Fukuda Stepping Test. Why you did both?

The authors have rewritten the manuscript differentiating the tests performed with accelerometers (indicated in the text as accelerometric functional tests) and the most frequently used clinical evaluation tests of balance (which we refer to in the text as clinical tests).

  1. Methods. Paragraph 2.5 does not make sense. What does “the review showed us what accelerometric tests were relevant…” mean? You defined before a lot of measurement and now you don’t know which use?

In this sentence, the authors mean that to decide which functional accelerometric tests and which clinical tests to apply, we previously carried out a systematic review.

The authors have added the bibliographic reference of this systematic review.

  1. Methods. Line 147: “y” is not English.

That typo has been corrected.

  1. Methods. Line 148: statistical analysis must be discussed on appropriate paragraph.

The authors have corrected the paragraph and moved the statistical information completely to the next subsection of Materials and Methods.

  1. Methods. Root means square must be introduced in the appropriate paragraph (2.3) not here.

The authors have introduced the RMS concept in the subsection that you have indicated.

  1. Methods. Line 159: please identify between brackets the “clinical indicators” and the “accelerometric variables” for clarity.

The authors have completed the sentence with the information you indicate.

  1. Methods. Lines 163-165: please also here identify variables, insert them between brackets.

The authors have completed the sentence with the information you indicate.

  1. Results: Line 183: “anteroposterior axis and the RMS”, what you mean? You did not analyze the RMS of the AP and ML direction? If not, what kind of measure you used for AP and ML?

The phrase refers to two variables: (1) the acceleration in the anteroposterior axis and (2) the RMS of the accelerations in the three axes.

The authors did not calculate the RMS of each axis. We calculated the RMS of the three axes.

In the new version of the manuscript, the accelerometric variables analyzed have been presented and defined more clearly from the beginning and we hope that the paragraph is sufficiently understandable.

  1. Results: Table 1: what kind of measure is g (accelerometric variables). Moreover, the difference between ALL and ???? for AP what does it means? Did you analyze male vs. female? If yes, why ALL vs. ??? means?

The unit of measurement of the accelerometer used is g (unit of gravity). This concept has been adequately presented in the Materials and Methods section in the description of the instrument used.

The placement of the asterisks in Table 1 has been corrected to adequately refer to the comparison made between the sexes (girls vs. boys).

  1. Discussion. Lines 247-254: I suggest to you, to discuss results between male and female, to read and cite the paper of Persiani et al., (2015). Laterality of stance during optic flow stimulation in male and female young adults. BioMed Research International, 2015.

The indicated reference has been analyzed and included in the Discussion.

  1. Discussion. Lines 255-267: I suggest to you, to discuss results about gait and posture, to read and cite the paper of Raffi et al., (2017). Angle of gaze and optic flow direction modulate body sway. Journal of Electromyography and Kinesiology, 35, 61-68.

The indicated reference has been analyzed and included in the Discussion.

Once again, thank you very much for the time spent and the interest shown in this work; as well as in the positive evaluations you have given of it.

Receive a warm greeting,

The authors.

Reviewer 2 Report

Validity and Reliability of a Tool for Accelerometric Assessment of Balance in Scholar Children

Manuscript ID: jcm-1010652

General Comments:

The study aimed to compare clinical balance tests with accelerometer data during balance tests. I have outlined my suggestions and edits as specific comments below. Some major issues include the below:

The clinical indicators are not fully presented and their relationship with the accelerometric data need to be explained better.

Also, why can’t accelerometer data be collected, analyzed and correlated when actually doing the clinical balance tests? The clinical tests include assessments of both static and dynamic balance. Why can’t accelerometer data be collected and analyzed during these clinical balance tests? Please discuss more on the rationale for comparing one group of balance test with another group of clinical balance tests.

The need for the study in the introduction and more importantly the justification of the observed results in the discussion needs to be strengthened.

Specific Comments:

Abstract:

Please present a structured abstract. The abstract is not written well and doesn’t convey any procedures or observed quantitative results and their rationale. More importantly, there are sentences are the just flat-out incomplete and sentences that doesn’t make any sense. See examples below.

Line 16 – Sentence is incomplete.

Line 22 – What is the rationale for this sentence?

Abstract needs to be re-written to convey the study done. The last sentence is also an over-statement given the results observed.

Introduction:

Line 38 – Does not make sense and is an oversimplification of postural control mechanism.

Line 46 – What is already advanced stages mean? What is this referring to?

Methods:

Line 77 – Please move information on accelerometer from 2.3 to here. Describe the equipment first followed by procedures and then data analysis.  

Line 81 – what is a corresponding measurement room?

Line 85 – mention the first two trials as being done in bilateral stance.

Line 159 – What type of correlation analysis?

Results:

Can you please briefly mention the strength of the correlation in the results. I understand that they are discussed in the discussion section, but having a directional result helps understanding better.

Discussion:

Lines 232-235 – Briefly explain the observed results here before progressing into the discussion section.

Line 247 – Don’t write a worse result. Just compare between boys and girls and saying girls had better postural stability or balance.

Author Response

Dear Editor and Reviewer of Journal of Clinical Medicine:

Thank you very much for your suggestions and contributions to improve the quality of the manuscript. Following your indications, we respond, point by point, to the reviewers' comments.

In the text, all the modified or added sentences have been written in red to facilitate the correction by the reviewers.

  1. The clinical indicators are not fully presented and their relationship with the accelerometric data need to be explained better.

The Introduction and Materials and Methods sections have been rewritten and expanded so that clinical tests and accelerometric functional tests are adequately presented and described.

  1. Also, why can’t accelerometer data be collected, analyzed and correlated when actually doing the clinical balance tests? The clinical tests include assessments of both static and dynamic balance. Why can’t accelerometer data be collected and analyzed during these clinical balance tests? Please discuss more on the rationale for comparing one group of balance test with another group of clinical balance tests.

The experimental possibility that the reviewer describes is valuable and feasible. However, the objective of the present investigation was to achieve the simplest possible but reliable, valid and sensitive evaluation protocol at the same time.

The functional accelerometric evaluation tests were chosen based on a previous systematic review (which has been included among the bibliographic references used).

  1. The need for the study in the introduction and more importantly the justification of the observed results in the discussion needs to be strengthened.

The Introduction and Discussion sections have been expanded.

  1. Abstract: Please present a structured abstract. The abstract is not written well and doesn’t convey any procedures or observed quantitative results and their rationale. More importantly, there are sentences are the just flat-out incomplete and sentences that doesn’t make any sense. See examples below:

Line 16 – Sentence is incomplete.

Line 22 – What is the rationale for this sentence?

The last sentence is also an over-statement given the results observed.

Abstract needs to be re-written to convey the study done.

The authors have rewritten the Abstract (we have not included headings because the instructions for authors of the Journal indicate this).

  1. Introduction: Line 38 – Does not make sense and is an oversimplification of postural control mechanism.

The indicated phrase has been removed.

  1. Introduction: Line 46 – What is already advanced stages mean? What is this referring to?

The sentence has been rewritten.

  1. Methods: Line 77 – Please move information on accelerometer from 2.3 to here. Describe the equipment first followed by procedures and then data analysis.

The order of the sections 2.2 and 2.3 has been reversed, as you indicate.

  1. Methods: Line 81 – what is a corresponding measurement room?

The sentence has been rewritten.

  1. Methods: Line 85 – mention the first two trials as being done in bilateral stance.

This detail has been added to the description.

  1. Methods: Line 159 – What type of correlation analysis?

This detail has been added to the description.

  1. Results: Can you please briefly mention the strength of the correlation in the results. I understand that they are discussed in the discussion section, but having a directional result helps understanding better.

The subsection in which the results of the correlation analysis are presented has been rewritten.

  1. Discussion: Lines 232-235 – Briefly explain the observed results here before progressing into the discussion section.

After the first introductory sentence, a sentence has been added that briefly describes the findings obtained.

  1. Discussion: Line 247 – Don’t write a worse result. Just compare between boys and girls and saying girls had better postural stability or balance.

The sentence has been rewritten.

Once again, thank you very much for the time spent and the interest shown in this work; as well as in the positive evaluations you have given of it.

Receive a warm greeting,

The authors.

Round 2

Reviewer 1 Report

I'd like to thanks all the authors for their effort in improving the manuscript, which I suggest for pubblication in its present form. 

Reviewer 2 Report

The authors have addressed the concerns raised and revised the manuscript.